# Liposomal Artificial Red Blood Cell-Based Carbon Monoxide Donor Is a Potent Renoprotectant against Cisplatin-Induced Acute Kidney Injury

**DOI:** 10.3390/pharmaceutics14010057

**Published:** 2021-12-27

**Authors:** Kazuaki Taguchi, Yuto Suzuki, Moeko Tsutsuura, Kana Hiraoka, Yuki Watabe, Yuki Enoki, Masaki Otagiri, Hiromi Sakai, Kazuaki Matsumoto

**Affiliations:** 1Division of Pharmacodynamics, Faculty of Pharmacy, Keio University, 1-5-30 Shibakoen, Minato-ku, Tokyo 105-8512, Japan; awacsyuto@keio.jp (Y.S.); moeko.candybonbon9825@keio.jp (M.T.); kana.n03nr@gmail.com (K.H.); ku.yw622@keio.jp (Y.W.); enoki-yk@pha.keio.ac.jp (Y.E.); matsumoto-kz@pha.keio.ac.jp (K.M.); 2Faculty of Pharmaceutical Sciences, Sojo University, 4-22-1 Ikeda, Nishi-ku, Kumamoto 860-0082, Japan; otagirim@ph.sojo-u.ac.jp; 3DDS Research Institute, Sojo University, 4-22-1 Ikeda, Nishi-ku, Kumamoto 860-0082, Japan; 4Department of Chemistry, Nara Medical University, Shijo-cho 840, Kashihara 634-8521, Japan; hirosakai@naramed-u.ac.jp

**Keywords:** cisplatin, nephrotoxicity, liposome, artificial blood, carbon monoxide, apoptosis, anti-tumor, cancer, hemoglobin

## Abstract

Cisplatin (CDDP) is an essential anti-tumor agent for chemotherapeutic regimens against various types of cancer. However, the progression of nephrotoxicity, which is the main adverse effect of CDDP, leads to discontinuation of CDDP chemotherapy. Therefore, development of a renoprotectant against CDDP-induced nephrotoxicity is crucial. Here, the potential of a carbon monoxide (CO)-loaded hemoglobin-vesicle (CO-HbV) as a renoprotectant for CDDP-induced nephrotoxicity was evaluated for its renoprotective effects against CDDP-induced nephrotoxicity, inhibitory effects on the anti-tumor activity of CDDP, and anti-tumor activity. In healthy mice, after pretreatment with either saline, HbV, or CO-HbV prior to CDDP administration, only the CO-HbV pretreatment group ameliorated the progression of CDDP-induced nephrotoxicity by suppressing apoptosis via caspase-3. In experiments using B16-F10 melanoma cells, the half-maximal inhibitory concentration of CDDP decreased with co-incubation with CO-HbV, owing to the anti-tumor activity of CO. CO-HbV pretreatment had no impact on the anti-tumor activity of CDDP in B16-F10 melanoma cell-bearing mice, which was consistent with the results of the cell experiment. Furthermore, CO-HbV pretreatment improved body growth and survival rates. In conclusion, CO-HbV pretreatment is a potent renoprotectant for CDDP-induced nephrotoxicity, allowing treatment with CDDP to be conducted without failure of cancer treatment.

## 1. Introduction

Cisplatin (CDDP) is the first platinum anti-tumor agent approved by the Food and Drug Administration. Although it has been known to contribute to the treatment of various types of cancer for nearly half a half-century, 15–35% of cancer patients with CDDP treatment develop severe nephrotoxicity [1,2]. According to the European and Japanese guidelines for the prevention of CDDP-induced nephrotoxicity, CDDP hydration by saline before, during, and after CDDP infusion is strongly recommended as a standard pharmaceutical intervention [3,4]. Despite these practices, cancer patients receiving CDDP treatment still suffer from nephrotoxicity [5]. Therefore, researchers are gaining interest in identifying renoprotectants to prevent CDDP-induced nephrotoxicity [6,7,8].

Accumulated scientific research exploring the bioactivity of carbon monoxide (CO) has demonstrated its therapeutic potential towards acute kidney injuries [9,10,11]. Concerning CDDP-induced nephrotoxicity, Yoon et al. reported that treatment with CO-releasing molecule (CORM-3) protected normal kidney epithelial cells from CDDP-induced cell death in vitro [12]. Furthermore, Tayem et al. reported that daily CORM-3 treatment ameliorated the progression of CDDP-induced nephrotoxicity in a rat model [13]. These facts suggest that CO is a promising renoprotectant against CDDP-induced nephrotoxicity.

Recently, attempts have been made to develop unique CO donors that utilize hemoglobin molecules as carriers [14,15]. This approach is based on the physiological characteristics of hemoglobin in natural red blood cells where it functions as a carrier of endogenous CO in mammals [16]. Hemoglobin-vesicles (HbV) have been developed as a cellular type of artificial red blood cell in which concentrated human hemoglobin is encapsulated into liposomes [17]. Based on the fact that HbVs have a similar structure and function as natural red blood cells, attempts were made to develop an HbV-based CO donor that loads CO into hemoglobin inside HbV (CO-HbV) [18]. To date, CO-HbV has been demonstrated to have favorable characteristics as a CO donor, such as its CO-releasing properties [18]. Furthermore, CO-HbV ameliorated the clinical manifestations of disorders such as systemic ischemia reperfusion [19], pulmonary fibrosis [20], colitis [21], acute pancreatitis [22,23], and obliterative bronchiolitis [24] in animal models after only one or two administrations. Therefore, we hypothesized that CO-HbV also can suppress CDDP-induced nephrotoxicity through one-time prophylactic administration.

This study aimed to evaluate the potential of CO-HbV as a renoprotectant against CDDP-induced nephrotoxicity. For this purpose, we investigated the potent prophylactic effect of CO-HbV using a CDDP mouse model. In addition, the targeting action of CO-HbV was investigated. Finally, considering the dual-natured characteristics (cytoprotective and anti-tumoral) of CO against tumor cells, the influence of CO-HbV on B16-F10 melanoma cells was further evaluated both in vitro and in vivo.

## 2. Materials and Methods

### 2.1. Preparation of CO-HbV

DeoxyHbV suspension in saline was ([Hb] = 10 g/dL) prepared by the kneading method, as reported previously [25]. The resulting suspension was bubbled gently with 100% pure CO gas for 5 min to convert it to CO-HbV. In this method, nearly 100% of hemoglobin inside CO-HbV is converted to carbonylhemoglobin without any changes in the physicochemical characteristics of deoxyHbV [24]. The diameter of CO-HbV was regulated to ca. 220 nm with lipid compositions of 1,2-dipalmitoyl-sn-glycero-3-phosphatidylcholine, cholesterol, 1,5-O-dihexadecyl-*N*-succinyl-L-glutamate, and 1,2-distearoyl-sn-glycero-3-phosphatidyl-ethanolamine-*N*-PEG5000 at a molar ratio of 5:4:0.9:0.03, respectively.

### 2.2. Animals

A1l experiments were performed according to the protocol approved by the Institutional Animal Care and Use Committee of Keio University (approval number: 19035). ICR mice (4-week-old, male, 17–19 g) and C57BL/6J mice (5 weeks old, male, 18–23 g) were purchased from Japan SLC, Inc. (Shizuoka, Japan). Before each experiment, all mice were housed in a conventional room to acclimatize for one week.

### 2.3. Renoprotective Effect of CO-HbV against CDDP-Induced Nephrotoxicity in Mice

ICR mice were randomly treated with either saline (10 mL/kg, *n* = 8), HbV (1000 mg Hb/kg, *n* = 8), or CO-HbV (1000 mg Hb/kg, *n* = 8) via the tail vein. At 24 h after test sample administration, 20 mg/kg of CDDP (Randa^®^ Injection, Nihon Kayaku, Tokyo, Japan) was intraperitoneally (i.p.) administered. Control mice (*n* = 8) received saline alternatives to CDDP without any pre-treatment. At 72 h after CDDP injection, all mice were sacrificed to harvest blood and kidneys. Plasma samples were ultracentrifuged to remove the remaining HbV and CO-HbV before the determination of creatinine and blood urea nitrogen (BUN) levels [26]. The right kidney was immediately frozen at –80 °C, and the left kidney was embedded in paraffin for histological examination. One saline-treated CDDP mouse model died before sample collection.

### 2.4. Determination of Biological Parameters

The plasma levels of creatinine and BUN were measured using a Fuji DRI-CHEM 7000i (Fujifilm, Tokyo, Japan).

### 2.5. Histological Examinations

The paraffin-embedded kidney sections were stained with periodic acid-Schiff (PAS) for morphologic analysis using the usual method. For apoptosis analysis, sections were treated with 4’, 6-diamidino-2-phenylindole (DAPI; Dojin Chemical, Kumamoto, Japan) and an in situ cell death detection kit fluorescein (Roche, Basel, Switzerland). The images were obtained using an all-in-one fluorescence microscope (Keyence Corp., Osaka, BZ-X700, Japan). Apoptotic cells were assessed by counting terminal deoxynucleotidyl transferase dUTP nick end labeling (TUNEL) and DAPI double-positive cells as the mean of five fields from randomly selected images of each kidney section.

### 2.6. Western Blotting Analysis

The right kidneys were homogenized in radioimmunoprecipitation assay (RIPA) buffer containing 1% protease inhibitor cocktail and centrifuged at 12,000 rpm for 10 min at 4 °C. The protein content in the supernatants was quantified using the BCA protein assay (TaKaRa BIO INC, Tokyo, Japan). After pretreatment by heating with sample buffer, samples containing 20 µg of protein were loaded onto 15% sodium dodecyl sulfate polyacrylamide gels, followed by electrophoretic transfer to polyvinylidene fluoride (PVDF) membranes. The membranes were blocked by treatment with 5% skim milk for 60 min at room temperature. After washing the membranes three times with PBS containing 0.05% Tween-20, the membranes were incubated with rabbit anti-caspase 3 primary antibody (cat#: c8487; Sigma-Aldrich, St. Louis, MO, USA), or β-actin antibody (cat#: 4967; Cell Signaling Technology, Beverly, MA, USA) for 60 min at room temperature. The membranes were then washed with PBS containing 0.05% Tween-20, and treated with anti-rabbit IgG and HRP-linked antibodies (cat#: 7074; Cell Signaling Technology, Beverly, MA, USA) for 60 min at room temperature. All band images were acquired using the LAS 4000 system (Fujifilm, Tokyo, Japan). All band intensities derived from cleaved caspase-3 and β-actin were analyzed using ImageJ.

### 2.7. Cell Viability Assay

B16-F10 melanoma cells (ATCC^®^ CRL-6475TM) were obtained from Summit Pharmaceuticals International Corporation (Tokyo, Japan). Cells were cultured in Dulbecco’s modified Eagle’s medium (DMEM) supplemented with 5% fetal bovine serum, 100 U/mL penicillin, and 100 µg/mL streptomycin at 37 °C in 5% CO_2_. B16-F10 melanoma cell suspension (100 µL) was added to each well of 96-well plates at a density of 5 × 10^4^ cells/mL and incubated overnight. After removing the medium, fresh DMEM containing different concentrations of CDDP (100 µL; 0–1000 µM) was added to each well. Fresh DMEM containing CO-HbV (100 µL; 0, 0.02, 0.2, 2, 20, and 200 µg Hb/mL) was then added before incubation for 24 h at 37 °C in a CO_2_ incubator. Cell viability was determined using cell counting reagent (Nacalai Tesque, Inc., Kyoto, Japan).

### 2.8. Influence of CO-HbV on Tumor Growth in B16-F10 Melanoma Cell-Bearing Mice

C57BL/6J mice (*n* = 40) were subcutaneously inoculated with B16-F10 melanoma cells (2 × 10^6^ cells/mouse). The experiment commenced (day 0) when tumor sizes were 50–100 mm^3^. All B16-F10 melanoma cell-bearing mice were treated with CDDP (10 mg/kg, i.p.) every 3 days (on days 1, 4, 7, and 10). One day before the CDDP treatment (on days 0, 3, 6, 9), B16-F10 melanoma cell-bearing mice received either saline (10 mL/kg, *n* = 10), HbV (1000 mg Hb/kg, *n* = 10) or CO-HbV (1000 mg Hb/kg, *n* = 10) via the tail vein. Body-weight and tumor volume were monitored daily for 14 days after the first CDDP treatment. The tumor volume was calculated by the following formula; tumor volume (mm^3^) = 0.5 × a (mm) × b^2^ (mm), where ‘a’ and ‘b’ were the largest and the smallest tumor diameter, respectively [27]. On day 14, surviving B16-F10 melanoma cell-bearing mice were euthanized by cervical dislocation.

### 2.9. Statistical Analysis

Student’s t-test or one-way analysis of variance (ANOVA) with Bonferroni correction were performed for comparisons. Results were considered statistically significant when the probability was less than 0.05. Cell viability assay was performed to determine the half-maximal inhibitory concentration (IC50) which was calculated based on the sigmoidal curve obtained using JMP (version 15) software (SAS Institute, Cary, NC, USA). Survival rates were analyzed using Kaplan–Meier curves and compared using the log-rank test.

## 3. Results and Discussion

### 3.1. Renoprotective Effect of CO-HbV on CDDP-Induced Nephrotoxicity

First, we investigated whether pretreatment with CO-HbV ameliorated the progression of CDDP-induced acute kidney injury in healthy mice. Mice were treated with either saline, HbV, or CO-HbV 24 h prior to CDDP administration. At 72 h after CDDP administration, the levels of BUN and creatine in plasma, which are biological parameters reflected the renal injuries, were significantly increased in the saline group (Figure 1A,B). In addition, severe damage to tubular (such as swelling and detachment) but not glomeruli were observed in the kidneys of saline-treated CDDP mice when compared to control mice (Figure 2A,B). This is consistent with the fact that CDDP induces toxicity by retaining proximal tubular epithelial cells [1]. In contrast, the clinical manifestations of CDDP-induced nephrotoxicity were significantly suppressed by pretreatment with CO-HbV (Figure 1 and Figure 2D) indicating that CO-HbV is a potent renoprotectant against CDDP-induced nephrotoxicity, consistent with the previous findings surrounding CORM-3 [13]. Notably, the HbV-treated CDDP mouse model showed similar results to that of saline pretreatment (Figure 1 and Figure 2C). This result suggests that CO liberation from CO-HbV is the major contributor to the renoprotection against CDDP-induced nephrotoxicity.

### 3.2. Protective Mechanism of CO-HbV against CDDP-Induced Nephrotoxicity

The pathogenesis of CDDP-induced nephrotoxicity is complex and involves several causative elements such as inflammatory cytokines, reactive oxygen species, and mitochondrial dysfunction, followed by the induction of apoptosis [28]. Thus, we next investigated the apoptotic cells in renal tissues of CDDP mouse model by TUNEL assay. The results revealed an increased number of TUNEL-positive cells in the CDDP mouse model treated with saline and HbV compared to control mice (Figure 3A–C,E). In contrast, the number of apoptotic cells was significantly decreased in the CO-HbV-treated CDDP mouse model (Figure 3D,E).

Previously, it was reported that CO acts anti-apoptosis against vascular muscular smooth cells and renal tubule epithelial cells by inhibiting the activation of caspase-3 [13,29]. Thus, the expression of cleaved caspase-3, an active form of caspase-3, was examined in the kidneys of the CDDP mouse model. Cleaved caspase-3 was significantly increased in the saline-treated CDDP mouse model (Figure 4). In contrast, the expression levels of cleaved caspase-3 in the kidney were decreased after CO-HbV pretreatment (Figure 4). Interestingly, HbV pretreatment also suppressed the activation of caspase-3 in the kidney when compared to saline pretreatment. The reason for this is unclear, but metabolites of HbV, such as bilirubin and CO, may be involved. It is known that CDDP-induced apoptosis in the kidney is mediated not only by caspase-3, but also by other effectors, such as p53 and high temperature requirement A2 (HtrA2/Omi) [7]. Therefore, HbV metabolites are likely not the only factors causing the strong suppression of multiple apoptotic pathways. Notably, CO-liberating CO-HbV has been shown to inhibit the activation of multiple apoptosis pathways, resulting in a strong suppression of apoptosis in the kidney. Moreover, CO gas molecules have also been reported to partly regulate apoptosis by suppressing p53 expression in vascular smooth muscle [30].

The association of the diverse physiological activity of CO with renoprotection against CDDP-induced nephrotoxicity involves several factors, such as interactions with reactive oxygen species and inflammation [28]. Several studies have investigated the curative or protective effects of anti-oxidative and anti-inflammatory agents against CDDP-induced nephrotoxicity [8,31,32]. Accumulated evidence indicates that the anti-inflammatory and anti-oxidative activities of CO reduce the onset and progression of organ injuries [33]. Although this study did not investigate whether the anti-inflammatory and anti-oxidative activity of CO was associated with the renoprotective effect of CO-HbV, these versatile biological activities of CO would comprehensively suppress CDDP-induced nephrotoxicity.

### 3.3. Effect of CO-HbV on Anti-Tumor Activity of CDDP against B16-F10 Melanoma Cells In Vitro

Considering the versatile cytoprotective effects of CO, it is important that the anti-tumor activity of CDDP should not be counteracted by co-administration of CO-HbV. In addition, many studies have reported the anti-tumor activity of CO by virtue of the anti-Warburg effect it has against tumors [34,35]. To assess this with respect to CO-HbV, the effect of CO-HbV on tumor cells was evaluated using B16-F10 melanoma cells. The viability of B16-F10 melanoma cells was slightly, but not significantly, decreased by the addition of CO-HbV (Figure 5A), suggesting the possibility of anti-tumor effects. Furthermore, the IC50 of CDDP, calculated from the dose-response curve shown in Figure 5B, was slightly decreased as the co-existing concentration of CO-HbV increased (IC50: 64.6, 69.5, 69.4, 58.7, 53.6, and 55.1 µM for 0, 1, 10, 100, 1000, and 10,000 µg Hb/mL of CO-HbV, respectively). Given these results, CO-HbV possesses weak anti-tumor activity rather than cytoprotective activity against B16-F10 melanoma cells.

### 3.4. Effect of CO-HbV on Anti-Tumor Activity of CDDP in B16-F10 Melanoma Cell-Bearing Mice

It was also examined whether CO-HbV pretreatment led to the successful anti-tumor activity of CDDP with minimal adverse effects in vivo. In this regard, the bodyweight change and survival rate in B16-F10 melanoma cell-bearing mice during repeated CDDP treatment after CO-HbV pretreatment were evaluated. The bodyweight of B16-F10 melanoma cell-bearing mice was decreased by repeated CDDP treatment (Figure 6A), and all mice died within 13 days (Figure 6B). Similar changes in body weight and survival were observed in B16-F10 melanoma cell-bearing mice pretreated with HbV prior to repeated CDDP administration (Figure 6A,B). On the other hand, CO-HbV pretreatment slightly reduced the body weight loss, and dramatically increased survival in comparison to saline- and HbV-pretreated CDDP mouse models (Figure 6A,B). Since all B16-F10 melanoma cell-bearing mice with no CDDP treatment survived without reduction in body weight during the observation period, the tumor did not cause a reduction in body weight and death. Although the extent of renal injury was not evaluated due to the insufficient number of surviving mice at the endpoint (on day 14), the amelioration of CDDP-induced nephrotoxicity by CO-HbV pretreatment would, in part, result in minimal impact on body weight and survival. Notably, the CDDP mouse model not only induces nephrotoxicity but also results in other adverse effects, such as gastrointestinal toxicity [36], and the protective roles of CO-HbV in such adverse events may contribute to its overall beneficial effects.

As for the effects of CO-HbV on the anti-tumor activity of CDDP, the tumor volume of the CO-HbV-pretreated CDDP mouse model was suppressed as much as that of the saline-treated CDDP mouse model (Figure 6C, left panel), indicating that CO-HbV did not interfere with the anti-tumor activity of CDDP in vivo. Interestingly, CO-HbV pretreatment slightly suppressed tumor growth compared to HbV pretreatment (Figure 6C, right panel). Previously, it was reported that HbV as an oxygen carrier enhanced irradiation therapy by tumor oxygenation [37], indicating that CO-HbV can deliver CO gas molecules to tumors. In addition, a weak anti-tumor activity of CO-HbV was observed in the cell experiments in the present study (Figure 5). Thus, CO delivery by CO-HbV to the tumor would at the very least contribute to the suppression of tumor growth. Fortunately, CO is toxic to cancer cells, but not to normal cell, due to the different response between normal cells and cancer cells after binding CO to cytochrome c oxidase [34]. This is the advantages of using CO therapy in cancer over other anti-tumor agents. Based on this fact, there is possibility that CO-HbV is developed as anti-tumor agent.

## 4. Conclusions

This study demonstrated that CO-HbVs exert a potent renoprotective effect against CDDP-induced nephrotoxicity without negatively affecting the anti-tumor activity of CDDP. Notably, CO-HbV can protect against CDDP-induced nephrotoxicity with a single administration before CDDP treatment; such reduction of frequency administration is the most significant advantage of CO-HbV over other CO donors. As of 2020, a phase 1 clinical trial of HbV as an artificial red blood cell has commenced [17,38], and it would be expected that this would translate into the practical application of CO-HbV as an HbV derivative in the future.

## Figures and Tables

**Figure 1 pharmaceutics-14-00057-f001:**
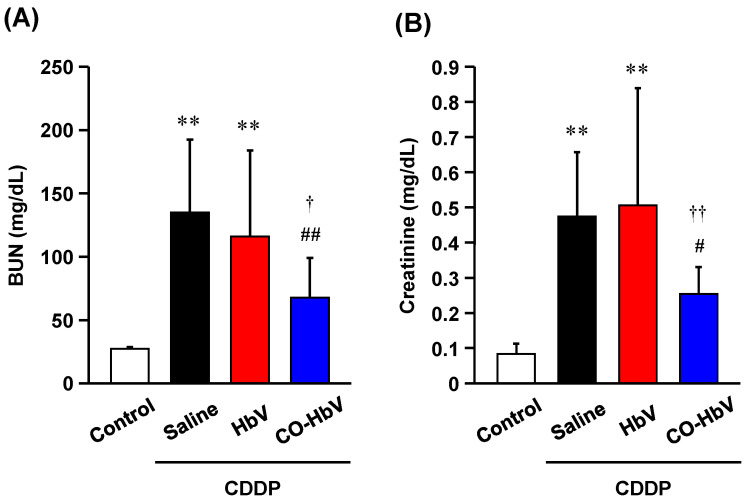
Biochemical analysis of kidney injury. The levels of (**A**) blood urea nitrogen (BUN) and (**B**) creatinine at 96 h after each sample treatment in cisplatin (CDDP) mice model. The data represent the mean ± standard deviation (S.D.). Control (*n* = 8), saline (*n* = 7), Hemoglobin-vesicle (HbV) (*n* = 8), carbon monoxide (CO)-loaded HbV (CO-HbV) (*n* = 8); ** *p* < 0.01 vs. control, # *p* < 0.05, ## *p* < 0.01 vs. Saline, † *p* < 0.05, †† *p* < 0.01 vs. HbV.

**Figure 2 pharmaceutics-14-00057-f002:**
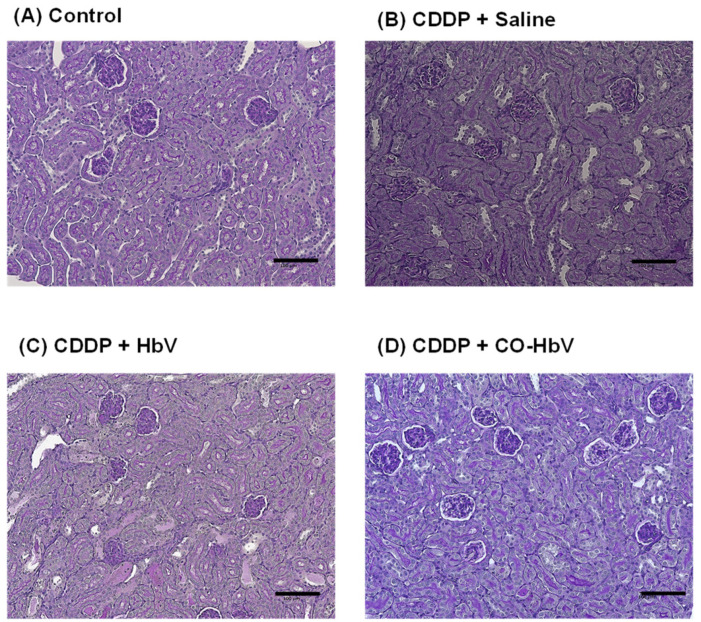
Representative imaging of kidney stained by PAS at 96 h after each sample treatment in cisplatin (CDDP) mouse model. (**A**) Control mice (without CDDP treatment), CDDP mouse model treated with (**B**) saline, (**C**) Hemoglobin-vesicle (HbV), and (**D**) carbon monoxide (CO)-loaded HbV (CO-HbV). Scale bar: 100 µm.

**Figure 3 pharmaceutics-14-00057-f003:**
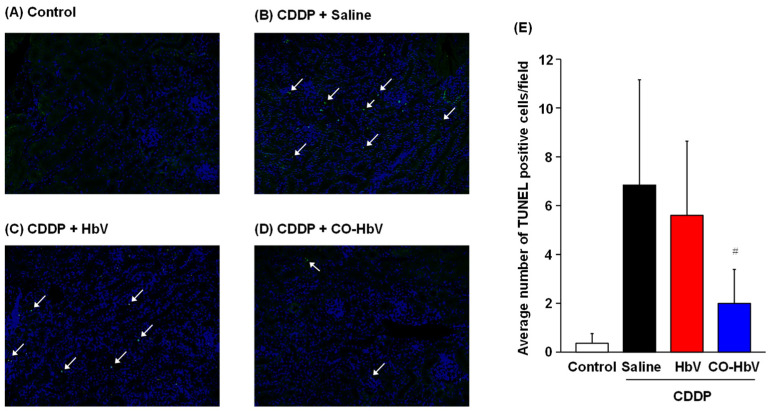
Representative TUNEL image of kidney at 96 h after each sample treatment in cisplatin (CDDP) mouse model. (**A**) Control mice (without CDDP treatment), CDDP mouse model treated with (**B**) saline, (**C**) Hemoglobin-vesicle (HbV), and (**D**) carbon monoxide (CO)-loaded HbV (CO-HbV). The images shown as merged images of TUNEL-positive cells (green) and DAPI staining (blue) (200× magnification). (**E**) Quantitative image analysis of TUNEL (green) and DAPI (blue) positive cells (white arrows). The data represents the mean ± S.D. Control (*n* = 8), Saline (*n* = 7), HbV (*n* = 8), CO-HbV (*n* = 8); # *p* < 0.05 vs. Saline.

**Figure 4 pharmaceutics-14-00057-f004:**
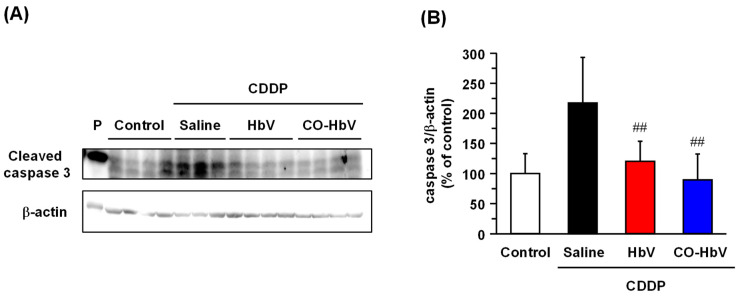
The effect of each treatment on the caspase-3 activation in kidney of cisplatin (CDDP) mouse model. (**A**) Representative protein expression and (**B**) quantitative analysis of band intensity of caspase-3 cleavage protein in kidney at 96 h after sample treatment. The data represents the mean ± S.D. Control (*n* = 8), Saline (*n* = 7), Hemoglobin-vesicle (HbV) (*n* = 8), carbon monoxide (CO)-loaded HbV (CO-HbV) (*n* = 8); ## *p* < 0.01 vs. Saline. P; positive control.

**Figure 5 pharmaceutics-14-00057-f005:**
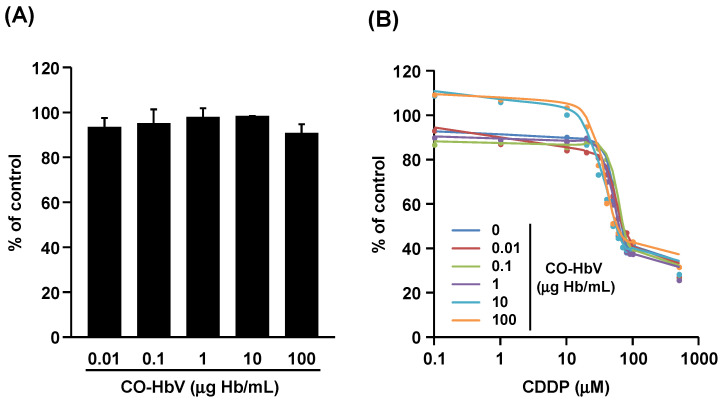
The effects of carbon monoxide (CO)-loaded Hemoglobin-vesicle (CO-HbV) on tumor cells. (**A**) Dose-dependent cytotoxicity effect of CO-HbV on B16-F10 melanoma cells. (**B**) The inhibitory effects of CO-HbV on anti-tumor activity of cisplatin (CDDP) toward B16-F10 melanoma cells. The data represent the mean ± S.D. (*n* = 3).

**Figure 6 pharmaceutics-14-00057-f006:**
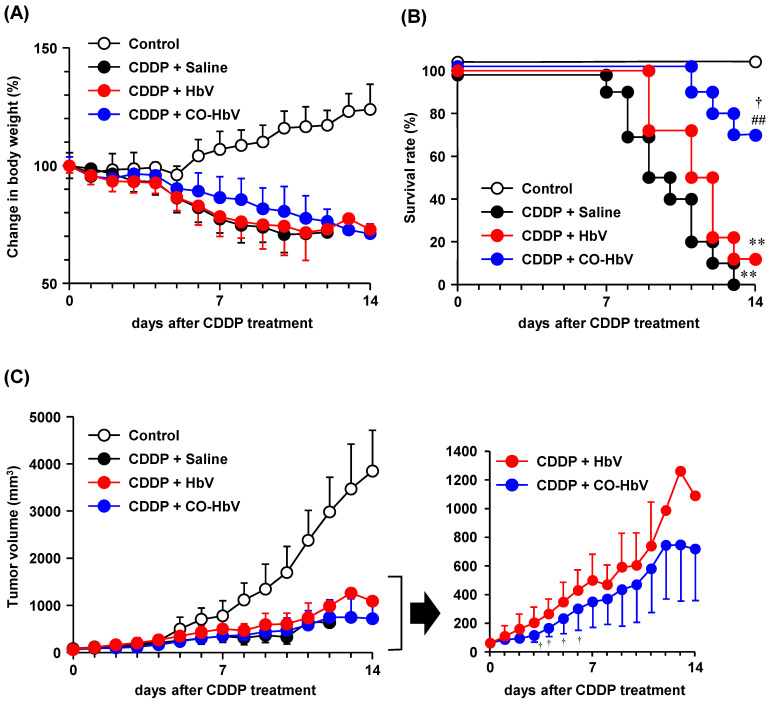
Effect of carbon monoxide (CO)-loaded Hemoglobin-vesicle (CO-HbV) on the anti-tumor efficacy of the conventional cisplatin (CDDP) treatment in B16-F10 bearing mice. The profiles of (**A**) body weight, (**B**) survival rate and (**C**) tumor volume until 14 days after the commencement of CDDP administrations. The data represent the mean ± S.D. (*n* = 10). ** *p* < 0.01 vs. Control, ## *p* < 0.01 vs. Saline, † *p* < 0.05 vs. HbV. The body weight differences of saline, HbV, and CO-HbV groups were significant (*p* < 0.01) from day 2 to day 10, from day 2 to day 11, and from day 2 to day 14, respectively, compared to the control group. The tumor volume of saline, HbV and CO-HbV group were significantly different (*p* < 0.01) from day 6 to day 10, from day 6 to day 11, and from day 6 to day 14, respectively, compared to the control group.

## Data Availability

Not applicable.

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
