# Peer review of "Liposomal Artificial Red Blood Cell-Based Carbon Monoxide Donor Is a Potent Renoprotectant against Cisplatin-Induced Acute Kidney Injury"

_pharmaceutics, 2021, doi:10.3390/pharmaceutics14010057_

Round 1
Reviewer 1 Report
The manuscript “Liposomal artificial red blood cell-based carbon monoxide donor is a potent renoprotectant against cisplatin-induced acute kidney injury” by Taguchi et al. explores the use of CO-HbV produced by the Sakai group for a new application. The manuscript was successful in showing the beneficial effects of CO-HbV in attenuating cisplatin toxicity, but we recommend the following comments be addressed prior to publication:
Major comments:
- Comment on the CO offloading kinetics of CO bound to Hb compared to other agents capable of releasing CO.
- Elaborate on the significance of Figure 2 in the manuscript. I’m not sure what it is that is different between the images and why it’s important.
- Is it known whether HbV or CO-HbV have any influence on the blood urea nitrogen, creatinine concentrations, or caspase-3 in the absence of CDDP?
- In Figure 3 (A) – (D) It seems as though the concentration/density of cells stained with DAPI in slides (A) and (D) are significantly lower compared to (B) (C), and therefore would have a lower concentration of green TUNEL positive cells. Comment on this?
- On lines 298-300, “…this is the most significant advantage of CO-HbV over other CO donors.” What exactly is the advantage? Does this mean other CO donors cannot protect against CDDP induced toxicity?
Author Response
Comment 1
Comment on the CO offloading kinetics of CO bound to Hb compared to other agents capable of releasing CO.
Reply 1
We have never compared the capable of CO releasing between CO-HbV and other CO donors directly. However, we previously evaluated the CO releasing from CO-HbV [1]. As a result, CO-HbV continuously released CO for a period of 8 h. Thus, CO-HbV possesses the capable of contentious CO releasing.
[1] Taguchi K, Nagao S, Maeda H, Yanagisawa H, Sakai H, Yamasaki K, Wakayama T, Watanabe H, Otagiri M, Maruyama T. Biomimetic carbon monoxide delivery based on hemoglobin vesicles ameliorates acute pancreatitis in mice via the regulation of macrophage and neutrophil activity. Drug Deliv. 2018 Nov;25(1):1266-1274.
Comment 2
Elaborate on the significance of Figure 2 in the manuscript. I’m not sure what it is that is different between the images and why it’s important.
Reply 2
The CDDP-induced nephrotoxicity is morphologically characterized by (1) tubular damages such as swelling and detachment and (2) no glomeruli damage. Images of kidneys stained by PAS (Figure 2) showed these structural morphology changes in saline-treated CDDP mice model, meaning that our mice model is adequate to investigate the therapeutic potential of CO-HbV against CDDP-induced nephrotoxicity. Therefore, these images of kidney as shown in Fig. 2 is important to corroborate the therapeutic efficacy of CO-HbV against CDDP-induced nephrotoxicity.
Comment 3
Is it known whether HbV or CO-HbV have any influence on the blood urea nitrogen, creatinine concentrations, or caspase-3 in the absence of CDDP?
Reply 3
In previous, we evaluated the safety of HbV and CO-HbV in healthy mouse and rats [1, 2]. As a result, the level of blood urea nitrogen and creatinine did not show any obvious changes after the administration of HbV (2000 mg Hb/kg) and CO-HbV (1000 mg Hb/kg). Based on these findings, HbV and CO-HbV administration would not influence on the blood urea nitrogen and creatinine concentration in present study. As for caspae-3, we have no data regarding the changes of caspase-3 cleavage after both HbV and CO-HbV administration in ant organs of healthy animal. Since we are currently studying how CO-HbV modify the caspase-3 pathway in other disorder models, we would like to evaluate the effects of HbV and CO-HbV on the caspae-3 pathway in healthy animal at the same time.
- Nagao S, Taguchi K, Miyazaki Y, Wakayama T, Chuang VT, Yamasaki K, Watanabe H, Sakai H, Otagiri M, Maruyama T. Evaluation of a new type of nano-sized carbon monoxide donor on treating mice with experimentally induced colitis. J Control Release. 2016 Jul 28;234:49-58.
- Sakai H, Horinouchi H, Masada Y, Takeoka S, Ikeda E, Takaori M, Kobayashi K, Tsuchida E. Metabolism of hemoglobin-vesicles (artificial oxygen carriers) and their influence on organ functions in a rat model. Biomaterials. 2004 Aug;25(18):4317-25.
Comment 4
In Figure 3 (A) – (D) It seems as though the concentration/density of cells stained with DAPI in slides (A) and (D) are significantly lower compared to (B) (C), and therefore would have a lower concentration of green TUNEL positive cells. Comment on this?
Reply 4
As reviewer mentioned, fluorescence area derived from DAPI in slide (A) and (D) may look lower than that of (B) and (C). However, we checked five fields from randomly selected images of each kidney section and fluorescence intensity from DAPI is almost same among groups. Thus, we think that there is no problem regarding the analysis of TUNEL positive cell.
Comment 5
On lines 298-300, “…this is the most significant advantage of CO-HbV over other CO donors.” What exactly is the advantage? Does this mean other CO donors cannot protect against CDDP induced toxicity?
Reply 5
It is favorable to reduce the frequency of administration for the use of CO as renoprotectant from the viewpoint of clinical use. Although both CO-HbV and other CO donors including CORMs can protect against CDDP induced nephrotoxicity, CORM-3 therapy required daily administration to protect kidneys from CDDP toxicity. On the other hand, this study clearly showed that CO-HbV protected kidneys from CDDP-derived nephrotoxicity by one-shot administration. Thus, we think that CO-HbV has advantage over other CO donors, including CORM-3, from the viewpoint of the frequency of administration. Therefore, we revised sentence in Conclusion section in the revised manuscript. (line 304-305)
Reviewer 2 Report
This is a well-organized and well-illustrated research paper, has an important clinical message, and should be of great interest to the readers. The paper focused on the development of carbon monoxide loaded hemoglobin vesicles for the alleviation of cisplatin induced nephrotoxicity. Paragraphing is concise and good, and the article consists of important clinical findings and deserve publication after some revisions listed below.
- Are there any benefits of using CO-Hbv over the usage of CORM-3 based CO for alleviation of nephrotoxicity (Citation 13 Tayem et al). If there are benefits, please describe them in detail in the introduction section.
- Please mention more details about the synthetic procedure employed for CO-Hbv synthesis in the materials section.
- In the materials section, the diameter of CO-Hbv is mentioned as 220nm. Is there any evidence for this claim (example: TEM or DLS characterization for size measurement)?
- Please mention the importance and the prime reason for determining plasma creatine and BUN levels in the discussion section.
- Please cite previous reports on the advantages of using CO therapy in cancer and its importance in the discussion section
Author Response
Comment 1
Are there any benefits of using CO-Hbv over the usage of CORM-3 based CO for alleviation of nephrotoxicity (Citation 13 Tayem et al). If there are benefits, please describe them in detail in the introduction section.
Reply 1
It is favorable to reduce the frequency of administration for CO donor to use as renoprotectant in clinical situation. However, CORM-3 therapy required daily administration to protect kidneys from CDDP toxicity (Citation 13 Tayem et al). On the other hand, this study clearly showed that CO-HbV achieved renoprotective effect by only single administration. This would be advantage over other CO donors, including CORM-3. Since the description regarding these facts is inadequate to mention in the introduction section, we mentioned such benefit of CO-HbV in Conclusion section. (line 304-305)
Comment 2
Please mention more details about the synthetic procedure employed for CO-Hbv synthesis in the materials section.
Reply 2
As reviewer suggested, we added more detail method about preparation procedure employed for CO-HbV in the materials section. (line 76-78)
Comment 3
In the materials section, the diameter of CO-Hbv is mentioned as 220nm. Is there any evidence for this claim (example: TEM or DLS characterization for size measurement)?
Reply 3
In previous report, we showed the DLS data for size measurement (Citation 24). Since we used same batch of CO-HbV with previous report (citation 24), we unfortunately cannot show DLS characterization in this manuscript.
Comment 4
Please mention the importance and the prime reason for determining plasma creatine and BUN levels in the discussion section.
Reply 4
The plasma creatinine and BUN are biological parameters reflected the renal injury (function). Thus, these are important parameters to evaluate the renoprotective effects of CO-HbV against CDDP-induced nephrotoxicity. We added this explanation in the revised manuscript. (line 171-172)
Comment 5
Please cite previous reports on the advantages of using CO therapy in cancer and its importance in the discussion section
Reply 5
As reviewer suggested, we briefly described the advantages of using CO therapy in cancer and added citation in the discussion section. (line 288-292, citation 34)
34 Zhou, Y.; Yu, W.; Cao, J.; Gao, H. Harnessing carbon monoxide-releasing platforms for cancer therapy. Biomaterials 2020, 255, 120193.